EMBO
*reports*

# The dynamics of loss of heterozygosity events in genomes

Abhishek Dutta[1] & Joseph Schacherer [ID] [1,2] ✉

## Abstract

**Genomic instability is a hallmark of tumorigenesis, yet it also plays an essential role in evolution. Large-scale population genomics studies have highlighted the importance of loss of heterozygosity (LOH) events, which have long been overlooked in the context of genetic diversity and instability. Among various types of genomic mutations, LOH events are the most common and affect a larger portion of the genome. They typically arise from recombination-mediated repair of double-strand breaks (DSBs) or from lesions that are processed into DSBs. LOH events are critical drivers of genetic diversity, enabling rapid phenotypic variation and contributing to tumorigenesis. Understanding the accumulation of LOH, along with its underlying mechanisms, distribution, and phenotypic consequences, is therefore crucial. In this review, we explore the spectrum of LOH events, their mechanisms, and their impact on fitness and phenotype, drawing insights from *Saccharomyces cerevisiae* to cancer. We also emphasize the role of LOH in genomic instability, disease, and genome evolution.**

**Keywords** Genome Instability; Loss of Heterozygosity; DNA Repair; Mutation Accumulation; *Saccharomyces cerevisiae*
**Subject Categories** Chromatin, Transcription & Genomics; DNA Replication, Recombination & Repair; Evolution & Ecology

## Introduction

Exploring the factors influencing genome evolution constitutes a fundamental challenge in genetics, with implications in diverse processes including speciation, biotechnological advancements, and human diseases. Cells employ a repertoire of processes to faithfully replicate and preserve their genomes. Despite this, pathways involved in replication, recombination and repair are error-prone, resulting in mutations that impact cellular fitness and phenotypic outcomes. These events can influence the size, organization, and expression levels of genes, and modify genetic interactions involved in recombination and sex (Lynch et al, 2006; Rifkin et al, 2005). A large emphasis has been laid on exploring the patterns of germline mutations, while somatic mutagenesis has been poorly explored. While somatic genome alterations are not inherited, they have

significant effects on the clonal lineage of cells. For instance, somatic mutations are known to drive cancer development and progression in humans, whereas in microbes, they play a crucial role in enabling adaptation to new environments (Hanahan and Weinberg, 2000; Payen et al, 2016; Venkataram et al, 2016).

The complex nature of mutagenesis gives rise to diverse mutation spectra (frequency and distribution), characterized by distinct types, rates, and distributions of single and multi-nucleotide mutations (SNMs and MNMs), short insertions and deletions (indels), large structural variants (SVs), copy number variations (CNVs), loss of heterozygosity (LOH), and aneuploidies (Zhu et al, 2014; Liu and Zhang, 2019; Charron et al, 2019; Sui et al, 2020; Loeillet et al, 2020; Dutta et al, 2021). Somatic mutations extend their impact beyond cancers, influencing human health with many implications, including aging and microbial pathogenesis for example. Large-scale analyses of cancer genomes have identified cancer-type and tissue-specific mutation spectra that have become instrumental in acquiring prognostic and therapeutic insights (Vogelstein et al, 2013; Alexandrov et al, 2020).

Previous study across various model organisms indicates that mutation rates vary depending on genome position (Hawk et al, 2005; Wolfe et al, 1989). Local variation in mutation rates, a critical driver of genomic evolution, has been associated with several underlying mechanisms. These include differences in base composition, local recombination rates, and gene density, as well as variations in transcriptional activity, DNA repair efficiency at different genomic loci, chromatin structure, nucleosome positioning, and replication timing (Matassi et al, 1999; Hardison et al, 2003). Understanding these factors provides insights into how genomic stability and diversity are maintained across different regions of the genome. Much of the mutational variation mentioned is likely to occur in repetitive regions of the genome, which are subject to less selection pressure and appear to be less stable (Teytelman et al, 2008). The use of more precise and comprehensive high-throughput methods for measuring mutation rate variation, along with advanced bioinformatics approaches, will improve our understanding of the *cis*- and *trans*-acting factors that influence mutation rates. This exploration will be crucial to better understand the evolution, organization, and progression of genome disease. Advances in high-throughput genome sequencing of large populations, coupled with laboratory evolutionary experiments, have significantly improved our understanding of these events at the species level, providing a deeper understanding of genetic diversity, evolutionary processes, and underlying mechanisms that shape genomes across populations (Box 1).

[1]Université de Strasbourg, CNRS, GMGM UMR 7156, Strasbourg, France. [2]Institut Universitaire de France (IUF), Paris, France. ✉E-mail: schacherer@unistra.fr

Box 1.   In need of answers

1. How do different genetic backgrounds influence genome-wide patterns of loss of heterozygosity accumulation?
2. What role do LOH events play in generating genetic variation within natural populations?
3. How do DNA repair and recombination processes influence the spectrum of accumulating LOH events, and what can these spectra reveal about the mutagenic potential of the underlying pathways?
4. To what extent do LOH events contribute to mutational buffering?
5. What implications do LOH events have for our understanding of genomic stability and the evolutionary pressures shaping genome structure?

Among various mutation types, loss of heterozygosity (LOH) events have garnered significant attention due to their high frequency, their impact on large portions of the genome, and their crucial role in influencing a wide range of phenotypes and fitness. LOH typically occurs through the exchange of genetic material between homologous chromosomes, primarily via mitotic recombination, along with other mechanisms (Symington et al, 2014; Sui et al, 2020).

The budding yeast *Saccharomyces cerevisiae* has become an invaluable model for studying LOH, providing insights that are often difficult to obtain from other organisms. Research in *S. cerevisiae* has greatly advanced our understanding of genome maintenance and integrity at the genetic level. While mitotic recombination is a predominant mechanism for spontaneous LOH in diploid yeast, it also exhibits all the types of LOH observed in tumors, enabling broader insights into genetic instability and cancer development. In this review, we explore the diversity of LOH accumulation in the genome, the mechanisms driving these events, and the resulting impacts on fitness and phenotype.

## Mechanisms driving loss of heterozygosity

LOH events can be categorized into two types: copy-loss LOH (CL-LOH) and copy-neutral LOH (CN-LOH) (Fig. 1). In CL-LOH, a cell that was initially heterozygous at a particular locus loses one of its two alleles due to the deletion of one allele. In contrast, CN-LOH involves the loss of one allele, but instead of deletion, the remaining allele is duplicated to maintain the copy number. Homologous recombination (HR) plays a dominant role in repairing DSBs, accounting for ~90% of DSB repair in yeast and about 50% in vegetatively dividing mammalian cells (Johnson and Jasin, 2001; Valencia et al, 2001). CN-LOH is by far the most common outcome of DSB repair across the genome, significantly contributing to genetic variation and structural alterations (Nichols et al, 2020; Sui et al, 2020).

### Double-strand break repair (DSBR)-mediated LOH

LOH may arise from inappropriate repair of DNA double-strand breaks (DSBs), triggered by local DNA lesions or replication fork collapse (Pâques and Haber, 1999; Esposito and Bruschi, 1993; Acuña et al, 1994; Hiraoka et al, 2000; McMurray and Gottschling, 2003; Barbera and Petes, 2006; Sui et al, 2020). In both yeast and human cells, multiple repair mechanisms compete to resolve DSBs, typically through two major pathways. Non-homologous end joining (NHEJ) directly rejoins broken chromosome ends, while homologous recombination (HR) uses an intact DNA template for repair (Chang et al, 2017). When the identical sister chromatid serves as a template, there are no genetic consequences (Pâques and Haber, 1999; Symington et al, 2014). However, in diploid cells, repair using the homologous chromosome can lead to LOH and alterations in chromosome structure (Fig. 1).

HR pathways generally function to repair DNA double-strand breaks (DSBs), the nature of lesions underlying spontaneous LOH is not always clear (Cannan and Pederson, 2016). The DSB repair (DSBR) model initially described in the early 1980s (Szostak et al, 1983), is now widely recognized as the most reasonable explanation for the link between crossing over and gene conversion during HR (Fig. 2). The hallmark of HR lies in its dependence on a homologous template to faithfully restore genetic information lost at the initial DNA lesion. Considering a DSB as an initiating lesion, HR begins with the 5'→3' resection of the DSB ends to expose single-stranded DNA (ssDNA), on to which the Rad51 recombinase loads. A Double Holliday junction (dHJ) intermediate is produced as the DSBs are processed (Bzymek et al, 2010). dHJs must be removed to segregate the recombinant duplexes, which can be done using a helicase and topoisomerase to produce only non-crossover (NCO) products (dissolution), or by endonucleolytic cleavage (resolution) (Symington et al, 2014). Crossovers (CO) are produced by cleaving the inner strands of one HJ and the outer strands of the other, whereas cutting the inner strands of both HJs gives non-crossover (NCO) products (Mehta and Haber, 2014; Ahuja et al, 2021). CO with gene conversion is less common during mitotic recombination than it is during meiotic recombination (Andersen et al, 2008).

The synthesis-dependent strand annealing (SDSA) and migrating D-loop models were developed to explain the lower incidence of CO-associated gene conversions during mitotic DSB repair (McMahill et al, 2007; Mehta and Haber, 2014; Piazza and Heyer, 2018; San-Segundo and Clemente-Blanco, 2020). The sequence around a DSB is replaced with a copy of a homologous template through SDSA, which keeps the flanking regions in their original form (Miura et al, 2012). According to the migrating D-loop model, the invading ssDNA is dissociated and annealed to the other end of the DSB following unrestricted DNA synthesis (Pâques and Haber, 1999). Both these mechanisms result only in NCO products.

Break-induced replication (BIR) is yet another exciting but poorly understood pathway that repairs one-ended double-strand breaks (DSBs) like those formed by replication collapse or telomere erosion (Malkova et al, 2001; Kraus et al, 2001). During BIR, a chromosomal fragment is lost because of a DSB. The invading strand displaces the homolog to create a migrating D-loop bubble (Pâques and Haber, 1999). Both leading and lagging strand synthesis take place to restore lost chromosome ends. BIR is also considered highly mutagenic because of the activity of Polδ, which can also increase the susceptibility to chromosomal rearrangements and hence, is suppressed if two broken DNA ends are present (Llorente et al, 2008; Malkova and Ira, 2013). Somatic repair outcomes mechanisms often exhibit a bias against COs to mitigate the detrimental effects associated with LOH and SVs (Nichols et al, 2020; Liu et al, 2011). This stems from the functioning of error-prone polymerases that typically display low processivity and are

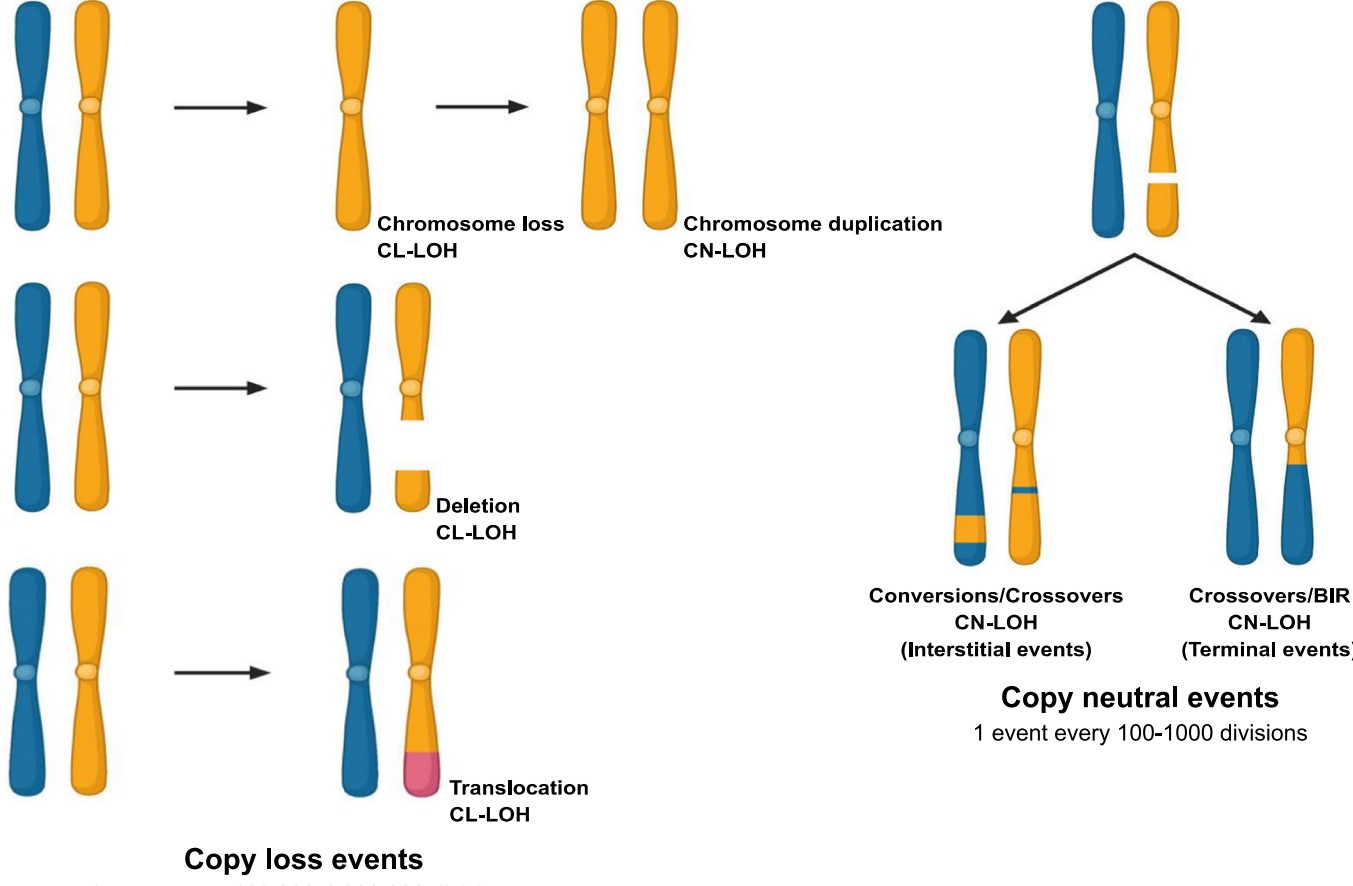

**Figure 1. Accumulation of LOH events.**

Potential mechanisms leading to LOH. LOH can span large regions of the chromosome through mitotic non-disjunction, with or without duplicating the remaining chromosome, Crossovers (CO) or Break-induced replication (BIR) events. Small deletions, translocations and gene conversions may result in short LOH events. The numbers represent the frequency of events in *S. cerevisiae* genomes [CL-copy loss; CN-copy neutral].

prone to slippage and stalling, facilitating the dissociation of the D-loop intermediate and subsequent invasion into homoeologous templates (Symington et al, 2014). This tendency towards frequent template switching contributes to elevated mutation rates and the generation of repair outcomes with distinctive mutational spectra at LOH sites. The observation of these mutational spectra in human genomes indicates that mechanisms implicated in DSB-induced mutagenesis in model systems may also function in humans (Loeillet et al, 2020).

DSBR-dependent CN-LOH outcomes are classified as interstitial or terminal LOH events based upon their positioning on chromosomes. In addition, they can also be discriminated depending upon the underlying repair pathways involved (Sui et al, 2020; Dutta et al, 2021). Interstitial events are outcomes of NCOs, double crossovers (DCOs) and COs that positioned away from chromosome ends, while terminal events are outcomes of BIR and COs involving chromosome ends. On average, interstitial events are short exchanges ranging from a few base pairs up to 10 kb in size, while in contrast, terminal LOH events frequently span large chromosomal regions larger than 100 kb, spanning the ends of chromosomes (Llorente et al, 2008; Malkova and Ira, 2013)

(Fig. 2). Apart from the differences in their chromosome-wide distributions and mechanisms involved, these LOH types have also been suggested to be outcomes of different initiating lesions (Sui et al, 2020). However, it is essential to note that not all terminal LOH events are reciprocal exchanges. NCOs/conversions frequent the terminal chromosomal regions and are promptly designated as terminal NCOs/conversions (Mancera et al, 2008; St. Charles et al, 2012; Yim et al, 2014; Laureau et al, 2016). Terminal LOH events due to conversions are significantly shorter in size relative to COs/BIR. Interestingly, terminal conversions are larger than interstitial LOH events (Laureau et al, 2016). LOH events genome-wide mainly comprise short interstitial events. The majority of LOH events in the genome are very short gene conversions (~100 bp or less on average), more than half of which impact only a single heterozygous position. These events contribute significantly to genome-wide homozygosity, comparable to the levels observed with reciprocal crossover (CO) events (Ene et al, 2018; Dutta et al, 2021). Having said that, it is also necessary to point out that such events are difficult to mechanistically dissect and detect unless done in unique model systems designed to detect them (Lee et al, 2009; St. Charles et al, 2012; Yim et al, 2014; Laureau et al, 2016).

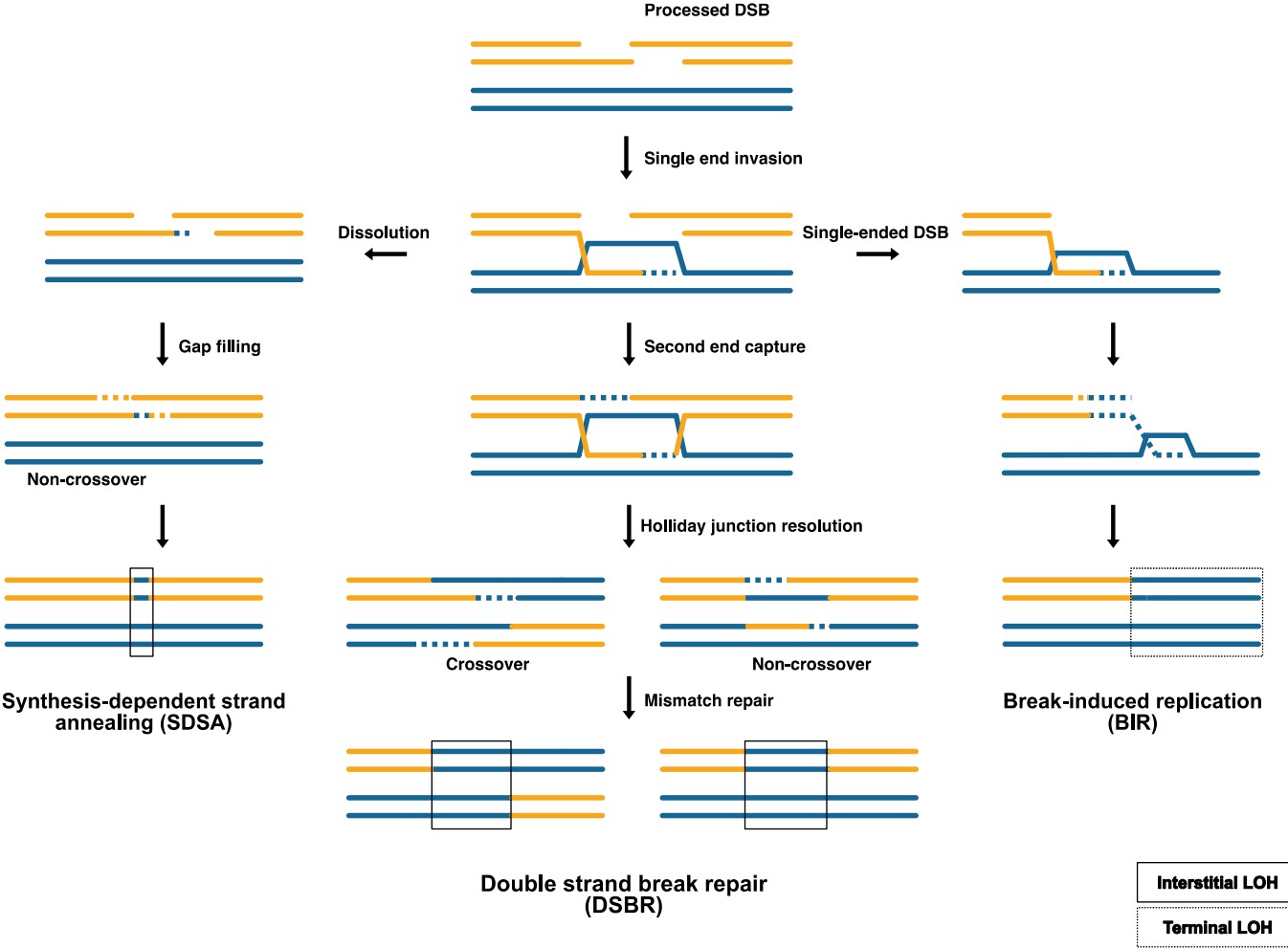

**Figure 2. Mitotic repair of DSBs by homologous recombination (HR).**

Recombination repair of a DSB is initiated by the 5′ resection of the break to produce single-stranded 3′ ends. The 3′ end invades the intact homolog and initiates leading strand synthesis. If a chromosome fragment is lost, repair is facilitated by Break-induced replication (BIR). The invading end creates a migrating D-loop, and both leading and lagging strand synthesis take place. The classical double-strand break repair (DSBR) pathway involves the further synthesis of the invading strand which gets ligated back to the resected 5′ end of the broken DNA molecule, leading to the formation of a double Holliday junction (dHJ). The dHJ can be resolved as crossovers and non-crossovers by Holliday junction resolvases. Repair by synthesis-dependent strand annealing (SDSA) is very common during mitotic DSB repair and involves the displacement of the invading strand with little synthesis and anneals to the other end of the broken DNA molecule, this is followed by gap repair and ligation leading to the formation of non-crossovers (adapted from Symington et al, 2014).

## Systemic instability

Traditional models of genome evolution typically assume that genomic alterations accumulate gradually and independently over time. Interestingly, a phenomenon known as mitotic systemic genomic instability (mitSGI) has been described in *S. cerevisiae*. Strains with an LOH event at a specific chromosomal location often display multiple unselected LOH events and mutations throughout their genome. This suggests that certain mitotic cells experience episodes of widespread genomic instability, making the entire genome more prone to further alterations over a short period of time (Sampaio et al, 2020). These results in the contexts of cancer and genomic disorders, challenge the notion of gradual accumulation of mutations (Gao et al, 2016; Liu et al, 2017a). While the specific mechanisms underlying this systemic instability in cells within a normal mitotic population remains unknown, Sampaio et al, proposed two likely scenarios: cellular aging and stochastic gene expression. First, replicatively aged yeast cells exhibit elevated rates of LOH, likely due to an impaired capacity to accurately detect and repair DNA double-strand breaks (McMurray and Gottschling, 2003). Second, variability in gene expression, a phenomenon observed across organisms ranging from prokaryotes to humans, may result in altered expression levels of genes involved in replication, recombination, and repair, thereby inducing a transient hyper-recombinogenic state (Liu et al, 2019). This systemic instability mirrors bursts of genomic instability seen in human cancers. For example, in genomic disorders characterized by multiple de novo CNVs, only one structural variant is typically causal, while the additional, unrelated CNVs arise during brief periods of instability. This underscores the potential importance of such events in the development of human disease (Liu et al, 2017b).

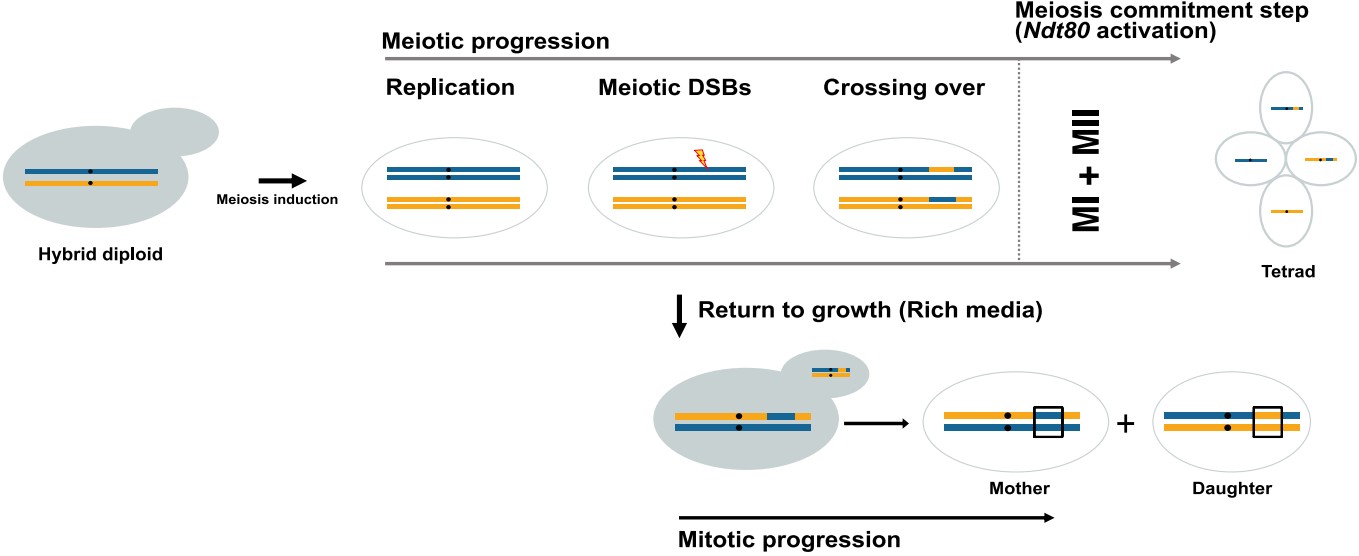

**Figure 3. Return-to-growth (meiotic abortions).**

A hybrid diploid cell is induced to initiate meiosis upon transfer to the sporulation medium, leading to Spo11-induced meiotic double-strand breaks (meiotic DSBs). If these cells are subsequently transferred to rich growth medium before the meiotic commitment step, the meiotic program is reversed. This process bypasses DNA replication and allows the meiotic mother cell to produce a diploid "daughter" cell through budding. Both the mother and daughter cells inherit two of the four chromatids present in the meiotic cell at the time RTG was induced, which may be recombined or non-recombined. The boxes highlight LOH events in the mother and daughter cells. The extent of LOH depends upon the time point of RTG induction.

## Meiotic abortions

Meiosis is a specialized cell division program in sexually reproducing organisms that reduces chromosome number by half to produce haploid gametes. This reduction is achieved through a single round of genome replication followed by two consecutive nuclear divisions without an intervening S phase. Although meiosis and mitosis share many similarities, several features are unique to meiosis. These include homolog pairing, synapsis, recombination, and the suppression of DNA replication before meiosis II, which are absent in the mitotic pathway (Pyatnitskaya et al, 2019). Unlike mitotic recombination, which occurs sporadically and unscheduled as part of the DNA damage response, meiotic recombination is a highly coordinated, systemic, and genome-wide process (Keeney et al, 2014). Mechanistic deviations from the standard meiotic program have been consistently observed across many organisms (d'Erfurth et al, 2009; Anderson et al, 2019; Hickman et al, 2013). A distinctive phenomenon observed in budding yeasts highlights the flexibility of the meiotic program, known as abortive meiosis or return-to-growth (RTG) (Esposito and Esposito, 1974; Dayani et al, 2011; Laureau et al, 2016; Mozzachiodi et al, 2021) (Fig. 3). Here, cells exit meiosis and re-enter mitotic division if transferred from sporulation to rich growth medium before committing to the meiosis I reductional division and undergo a mitotic-like equational chromosome segregation (Mozzachiodi et al, 2021; Laureau et al, 2016; Dayani et al, 2011). As a result, cells produce a mother-daughter pair that maintains the parental ploidy but exhibits recombined genotypes. Meiotic abortions involve rapid degradation of meiotic chromosomal structures and repairing recombination intermediates with minimal CO formation (Dayani et al, 2011). However, once meiotic commitment, i.e., *Ndt80*

activation is achieved, cells will complete meiosis regardless of a change in external cues. In *S. cerevisiae*, *Ndt80* is essential for the progression of meiosis, and disruptions result in normal levels of recombination but lead to pachytene arrest at the end of prophase I (Chu and Herskowitz, 1998). The commitment to meiosis is extremely sensitive to *Ndt80* dosage, and when the abundance is reduced, the meiotic commitment point shifts such that even cells undergoing the meiosis I division will exit the meiotic program and initiate mitotic cell division (Tsuchiya et al, 2014). A large-scale population genomics and life cycle characterization of over 1000 *S. cerevisiae* strains revealed widespread losses of sexual reproduction accompanied by pervasive genome-wide LOH across many subpopulations, highlighting the significant role of meiotic abortions in shaping the genomic landscape across many yeast species (Peter et al, 2018; De Chiara et al, 2022; Dutreux et al, 2023; Dutta et al, 2024).

## The rates of LOH accumulation

Population genomic surveys and mutation accumulation studies have consistently shown that LOH events are frequent, both in controlled laboratory settings and in natural populations. To understand the phenotypic and fitness consequences of these events, we can turn to the growing body of literature from experimental evolution. The most unbiased method of studying the rate and spectrum of genome-wide spontaneous mutations in the absence of selection is mutation accumulation (MA) experiments followed by whole-genome sequencing; however, some estimates have also been derived from experimental evolution experiments (Lynch et al, 2008, 2016; Dutta et al, 2017; Ene et al, 2018). Typical

laboratory evolution experiments (mutation accumulation and experimental evolution) utilize heterozygous yeasts propagated asexually, but a few experiments have also incorporated a sexual cycle (Nishant et al, 2010; Burke et al, 2014; Dutta et al, 2017; Liu and Zhang, 2021; McDonald et al, 2016).

MA lines undergo repeated single-colony bottlenecks over many generations, a process that allows for the detection and quantification of most mutations, including those that are mildly deleterious. In *S. cerevisiae*, MA experiments have been performed for 1700–4800 cell divisions in order to estimate mutation rates across different homozygous and heterozygous isolates involving laboratory and natural isolates with varying heterozygosity and ploidy levels (Nishant et al, 2010; Zhu et al, 2014; Dutta et al, 2017; Sui et al, 2020; Loeillet et al, 2020; Dutta et al, 2021, 2022). In addition, yeast MA lines have been propagated through varying environments and mutator genotypes (Liu and Zhang, 2019; Loeillet et al, 2020).

The frequency of LOH events is remarkably high, ranging from 0.3 to $5.6 \times 10^{-2}$ per cell division for interstitial LOH and 1.4 to $9.3 \times 10^{-3}$ per cell division for terminal LOH (Sui et al, 2020; Dutta et al, 2021). This corresponds to an average LOH rate ranging between 2.6 and $7.1 \times 10^{-5}$ per SNP per cell division, significantly exceeding the rate of point mutations, which typically ranges from 1 to $3 \times 10^{-10}$ per base pair per cell division for diploids (Sharp et al, 2018; Dutta et al, 2017, 2021; Zhu et al, 2014; Sui et al, 2020). The distribution patterns of interstitial LOH and terminal LOH are distinct from each other and from meiotic-associated gene conversions and crossovers. Terminal LOH events tend to be concentrated near telomeres, while interstitial LOH events are relatively evenly distributed, indicating potential differences in either the formation or the resolution of these events (Sui et al, 2020). The rates of LOH are influenced by various factors such as genetic background, level of heterozygosity, ploidy, and genomic region (Pankajam et al, 2020; Sui et al, 2020; Dutta et al, 2021, 2022; Tutaj et al, 2022). Certain genomic regions are particularly susceptible to LOH, notably the *S. cerevisiae* rDNA locus on chromosome XII, where SNPs near the telomere exhibit a LOH rate of $1.6 \times 10^{-4}$ per cell division (Peter et al, 2018; Sui et al, 2020). The extent of genome affected by LOH can vary significantly. In an MA experiment, an average of 15.9% of the genome underwent LOH, with some lines experiencing almost complete genome-wide LOH (Dutta et al, 2021). Although many mutation accumulation studies have focused on *S. cerevisiae*, rates of LOH have also been observed in MA studies of other asexual species. In yeasts belonging to the *Saccharomycodaceae* family, LOH rates range from 2 to $11 \times 10^{-6}$ per SNP per cell division (Nguyen et al, 2020). Large variations in LOH rates have also been noted in MA experiments performed in different environmental stresses including heat shock, nutrient limitation, hypoxia, and oxidative stress, highlighting the role of LOH in facilitating adaptations to new environments (Liu and Zhang, 2019). In addition, an MA study involving mutations in genome stability genes highlight the role of different DNA repair pathways in the generation of the LOH spectrum (Loeillet et al, 2020).

## LOH in polyploids

Polyploidy is a condition in which organisms have more than two complete sets of chromosomes (referred to as subgenomes, when the polyploid arises from the merging of non-identical genomes), typically resulting from whole-genome duplication (WGD) events. Newly formed polyploids often face challenges like chromosome missegregation and genetic instability (Comai, 2005; Otto, 2007; Van de Peer et al, 2017). As these polyploid genomes evolve, they frequently lose orthologous regions from one or all of the contributing subgenomes through mechanisms such as LOH, concerted chromosome loss, or rediploidization (Yoo et al, 2014; Lien et al, 2016; Du et al, 2020). In *S. cerevisiae* mutation accumulation experiments, LOH rates are markedly higher in polyploids, with triploids showing ~$2.2 \times 10^{-2}$ events per cell division and tetraploids around $8.4 \times 10^{-2}$. These LOH events often involve short, interstitial regions of the genome (Dutta et al, 2022). LOH has been shown to play a crucial role in shaping the structure and dynamics of the natural polyploid genomes of yeast species such as *Saccharomyces cerevisiae* and *Brettanomyces bruxellensis* (Peter et al, 2018; Eberlein et al, 2021).

Polyploid genomes have been demonstrated to be more sensitive to DSBs (Skonecznia et al, 2015). The high rates of LOH events in polyploid lines may be explained by the fact that heterozygous polyploid genomes may be associated with high DNA damage, which upon recombination repair results in increased LOH events. LOH in polyploids is typically asymmetric, indicating that losing alleles from the less expressed subgenomes may be less harmful and favored by natural selection (Alexander-Webber et al, 2016; Yoo et al, 2014). In addition, incompatibilities within the genomic context can further complicate retention and loss, influencing the evolutionary dynamics of polyploid organisms (Runemark et al, 2018).

## LOH as a driver of phenotypic variation and adaptation

Population genomic studies highlight the prevalence of hybridization events, often leading to a rapid expansion of phenotypic diversity. However, new hybrids frequently suffer from reduced fitness due to genetic incompatibilities between parental alleles, as many allelic combinations have not been fine-tuned by natural selection (Kondrashov et al, 2002; Greig et al, 2002; Morales and Dujon, 2012). LOH plays a pivotal role in stabilizing hybrid genomes, both within and between species, by purging incompatible allelic combinations. This process, observed in both natural and laboratory hybrids, is essential for adaptation, niche expansion, and resilience under stress (Payen et al, 2016; Lancaster et al, 2019; Smukowski Heil et al, 2017). In addition, hybridization events can trigger genome doubling, resulting in maladapted and unstable polyploids. These unstable polyploids often undergo genome reduction and LOH as a means of restoring stability (Marsit et al, 2021).

Microbial populations exhibit diverse phenotypes that enhance adaptability and fitness, influencing pathogenesis, persistence, and drug resistance. While SNPs were previously considered the primary source of this diversity, recent studies have highlighted the significant role of larger genomic alterations, such as LOH, structural variations (SVs), and copy number variations (CNVs). LOH, in particular, is crucial for fungal pathogenicity and drug resistance by generating extensive genomic variation. In *Cryptococcus* hybrids, LOH enhances pathogenicity (Li et al, 2012), while

in *S. cerevisiae* and *C. albicans*, LOH underpins adaptation to environmental stresses, including antifungal resistance (Cowen et al, 2015; Selmecki et al, 2010). Frequent LOH events near regions associated with drug resistance in *C. albicans* have been observed both in vivo and in vitro, highlighting their role in survival under antifungal pressure (Forche et al, 2008; Ford et al, 2015). CRISPR-mediated LOH serves as an effective tool for investigating the fitness impacts of genetic alterations in *S. cerevisiae*, demonstrating significant effects on fitness associated with LOH in the *ENA* and *MAL31* genes (Sadhu et al, 2016; James et al, 2019).

LOH can be an adaptive mechanism, particularly in diploid populations, by unmasking recessive or partially recessive beneficial mutations. Unlike dominant mutations, which are immediately subject to selection, recessive mutations often face barriers to fixation due to their lack of expression in the heterozygous state (Haldane's sieve). However, LOH bypasses this limitation by rendering these mutations homozygous (Orr and Betancourt, 2001). For instance, in *S. cerevisiae*, resistance to the antifungal drug nystatin was achieved through LOH events that uncovered recessive beneficial mutations, allowing them to spread rapidly throughout the population (Gerstein et al, 2014). Moreover, high LOH rates can match or even exceed the fixation rate of beneficial mutations in sexual populations (Mandegar and Otto, 2007).

Conversely, recessive deleterious mutations can limit LOH because unmasking these harmful mutations through LOH often result in reduced fitness or lethality. As a result, LOH is less likely to occur in regions containing recessive deleterious alleles, helping to preserve heterozygosity in these areas and ensuring the survival of the population. This was demonstrated in a long-term evolution experiment with *S. cerevisiae*; high-impact mutations in essential genes were often heterozygous in diploids, suggesting that LOH is less likely in these regions to preserve heterozygosity and viability (Johnson et al, 2021). Wild yeast strains exhibit high spore viability, highlighting the rarity of recessive lethal mutations. On the other hand, domesticated yeast, many of which have lost the ability to undergo meiosis, accumulate deleterious alleles (De Chiara et al, 2022). These alleles impose a genetic limitation, influencing the frequency and nature of LOH occurrences within the species, and thereby affecting its evolutionary trajectory. Wild isolates of *C. albicans* show high heterozygosity, making them an interesting subject for further study (Bensasson et al, 2019).

These examples highlight the role of LOH in increasing genetic variation, driving the fixation of beneficial mutations, and revealing gene–environment interactions. Collectively, they illustrate how LOH shapes phenotypic diversity and influences evolutionary trajectories.

## LOH in the cancer context

Various somatic genetic and epigenetic processes contribute to cancer development, including copy number alterations, deletions, gene rearrangements or translocations, somatic point mutations, and promoter hypermethylation (Garraway and Lander, 2013). LOH and allelic imbalance are widespread in human cancers, where individual cells can have several thousand regions under LOH. This process plays a key role in cancer progression by changing the levels of gene expression that are genetically or epigenetically altered (Feinberg and Tycko, 2004). LOH events expedite cancer-specific vulnerabilities by eliminating genetic redundancies. Several genes have been shown to be under LOH in cancers, leading to significant variability in the genotypic landscape of tumor and healthy cells (Nichols et al, 2020; Mansouri et al, 2022). LOH is frequently associated with a reduction in the copy number of the wild-type allele, which can result in a genotype becoming dominant as the wild-type allele loses function, a mechanism prevalent in many cancer types (Lengauer et al, 1998). These events are in fact a key hallmark of cancer, first identified through the use of polymorphic markers that were heterozygous in germline DNA but homozygous in tumor cells (Cavenee et al, 1983). Tumor suppressor genes regulate cell division and promote programmed cell death, at appropriate times. When these genes malfunction, cells can proliferate uncontrollably, increasing the risk of cancer development. Knudson's two-hit hypothesis has been pivotal in elucidating the role of tumor suppressor genes and the mechanisms underlying familial tumor-predisposing syndromes (Knudson, 1971). Therefore, for a cell to become cancerous, both of its tumor suppressor genes must be mutated. Simply, Knudson's two-hit hypothesis posits that the first "hit" occurs in the germline, affecting a tumor suppressor gene, while the second "hit" occurs somatically in the other allele of the same gene. CN-LOH is the most prevalent mechanism for second allele inactivation. A well-known example is the inactivation of both alleles of the retinoblastoma gene (Rb), which established it as one of the first recognized tumor suppressors (Cavenee et al, 1983; Murphree and Benedict, 1984). This discovery paved the way for identifying other tumor suppressors by analyzing regions with frequent LOH in human cancers. The majority of *BRCA1*-related breast and ovarian cancers, including up to 93% of ovarian cancers with *BRCA1* mutations, have been linked to LOH events (Maxwell et al, 2017).

Epigenetic modifications, such as DNA methylation and histone modification, often work in tandem with LOH to drive tumorigenesis. For example, promoter hypermethylation can silence the expression of the remaining functional allele of a tumor suppressor gene, even if no mutation is present. This creates a scenario where both genetic (LOH) and epigenetic mechanisms contribute to the loss of tumor suppressor gene function, accelerating cancer progression. CN-LOH regions are frequently associated with increased promoter methylation, which can silence gene expression without altering the DNA sequence. This epigenetic silencing can further compromise gene function in critical pathways, compounding the effects of LOH. In regions of LOH involving copy losses, gene expression is often reduced due to haploinsufficiency, impairing the ability to maintain normal regulatory functions, as a single functional copy of the gene may not produce sufficient support for normal cellular processes (Esteller et al, 2001; Ryland et al, 2015).

Advances in genomics have shown that LOH affects not only tumor suppressor genes but also nearby non-driver genes (Pedersen and De, 2013). In fact, LOH of non-driver genes far exceeds that of the limited oncogenes and tumor suppressor genes across various cancer types, generating renewed interest in their roles in cancer development and potential therapeutic applications (Ryland et al, 2015; Nichols et al, 2020). Thus, a systematic assessment of potential targets that integrates genome-wide evaluations of non-driver genes, variations in human genomes, and rates of LOH across various cancers is a crucial next step for future work.

## Perspectives

Mitotic recombination during somatic propagation is very common leading to elevated mutagenesis, primarily loss of heterozygosity. Nevertheless, the relative contributions of DNA replication and repair errors to mutation rates and LOH are still an open question in evolutionary biology. Stress or environment-induced mutagenesis has been explored in microbial evolution as well as in cancer (Maharjan and Ferenci, 2017). In contrast, the genetic background effect has been less explored. Importantly, in humans, population-specific differences in mutation spectra have been reported (Harris and Pritchard, 2017). Similar variability has been observed in yeast, *Drosophila*, mice and even cancers (Alexandrov et al, 2020; Dumont, 2019; Jiang et al, 2021).

The effect of genetic backgrounds is well established across a range of biological processes, however, their role in the context of mutation rate and spectra are poorly understood. It has been hypothesized that the variation in the mutation accumulation spectra may arise from the differences in the frequency of DNA lesions (DSBs) and/or differential processing of similar lesions based upon the genetic background (Sui et al, 2020). Genome sequencing and mutational landscape analyses of germline and somatic mutations have facilitated the identification of environmental sources of mutagen exposures (Pleasance et al, 2010). Unfortunately, the eventual mutation spectra and landscape can only act as a cumulative record of several mutational processes in the cell. Therefore, retrospectively determining the mechanistic origins of mutations remains a significant challenge. Understanding the regulation of these pathways may lie in their mutation spectra. Moreover, the mutational spectra shaped by DNA damage and repair pathways have undergone limited characterization, leading to a poor understanding of the mutagenic potential of these processes.

Genetic interactions can vary significantly across backgrounds, and while the mechanisms underlying these variations remain unclear in many cases, high-throughput experiments are shedding light on how genetic background influences functional variation within a species (Mullis et al, 2018). Laboratory model organisms have been pivotal for understanding core biological functions, but their narrow genetic scope has often overlooked the natural variation present in large populations. Incorporating this diversity is essential for uncovering the genetic basis of molecular and phenotypic variation and for deepening our understanding of complex biological mechanisms.

Another important aspect to consider is the dynamics of recombination rate variation and its impact on mutational load. Population genetic analyses in humans, *Drosophila*, and several other model organisms have shown that genomic regions with low recombination rates tend to accumulate a higher load of deleterious mutations (Bachtrog and Charlesworth, 2002; Presgraves, 2005; Neiman and Taylor, 2009). Investigating this further could enhance our understanding of mutational buffering and genome organization, particularly in terms of the spatial arrangement of critical genes in regions with varying recombination rates. Further exploration could provide insights into the genomic features associated with mutational hotspots and help explain how human populations manage to survive despite high mutation rates (Reed and Aquadro, 2006). By understanding the relationship between recombination rates and mutation load, we can better grasp the mechanisms of genomic stability and the evolutionary pressures shaping our genomes. This knowledge is crucial for identifying regions of the genome that are more susceptible to mutations and for developing strategies to mitigate the impact of these mutations on health and disease.

## Peer review information

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

## Acknowledgements

This work was supported by the European Research Council (ERC Consolidator Grant 772505). JS is a Fellow of the University of Strasbourg Institute for Advanced Study (USIAS) and a member of the Institut Universitaire de France.

## Author contributions

**Abhishek Dutta**: Writing—original draft; Writing—review and editing. **Joseph Schacherer**: Writing—original draft; Writing—review and editing.

## Disclosure and competing interests statement

The authors declare no competing interests.

