## [Peer Review File · EMBO Reports]

The dynamics of loss of heterozygosity events in genomes

Abhishek Dutta and Joseph Schacherer

Corresponding author(s): Joseph Schacherer (schacherer@unistra.fr)

Review Timeline:

Submission Date:	2nd Sep 24
Editorial Decision:	4th Oct 24
Revision Received:	18th Nov 24
Accepted:	9th Dec 24

Editor: Esther Schnapp

Transaction Report:

Dear Joseph,

Thank you for the submission of your Review to EMBO reports. We have now received the enclosed reports on it.

As you will see, all referees find the review topic interesting and a good fit for our journal. They also have several suggestions for how the review could be strengthened, and I think all suggestions are good and should be addressed, but please let me know in case you disagree. Please also submit a detailed point-by-point response with your final review.

I also discussed your review with my colleagues and we would like to suggest that more data on LOH in humans could be added. LOH is relevant and known to occur in humans (where it can unmask recessive mutant alleles at high rates and thus lead to pathogenesis), and we think it would be interesting to cover this aspect in more detail, may be in a separate section of the review, to make it more appealing to a wider readership.

We also think that it would be interesting to cover a more comprehensive diversity of organisms to which LOH might be relevant.

Finally, I would like to ask you to add a Box to the review called "In need of answers" that lists open questions in the field, which can be accompanied by suggestions for how these questions could be addressed experimentally.

As for timing, would it be possible for you to submit the revised review by November 4th? If you anticipate a problem in meeting this date, then please just let me know.

We will most likely not be able to include your review in our December issue, as the deadline is in early November. Our graphics designer also needs 10 days to redraw your figures. However, if you can submit the revised review at the end of October and if you can provide high quality figures that can be used as they are without involving a graphics designer, then we might be able to meet the deadline for our December issue. Both options, including the review in either our December or January issue, are fine with me.

With best wishes,
Esther

Referee #1:

Loss of Heterozygosity events have been a research focus for many decades, and their existence was hypothesized before their observation. Recently, there is increased interest in LOH events, in part because of their importance in experimental evolution studies such as those discussed in the review. The authors nicely connect mechanistic understanding of the LOH events with observations of their occurrence and evolutionary consequences. Other recent reviews have also noted these connections as well (see <https://www.ncbi.nlm.nih.gov/pmc/articles/PMC10276065/>), albeit with a somewhat different focus and some different references. This review could potentially profit from incorporating some references. In particular, additional studies of adaptation following LOH would strengthen the punch line of the paper.

I am somewhat concerned about the readability of some of the sections. In the section "Processes underlying loss of heterozygosity events", the second paragraph continues for a long time without breaks. This section needs to be broken up into more digestible chunks (both in paragraph structure and flow).

Additional comments:

The authors provide a review of LOH mechanisms, including homologous recombination (HR), non-homologous end joining (NHEJ), and break-induced replication (BIR). It also incorporates few experimental evolution studies, natural population genomics surveys, and laboratory-controlled mutation accumulation (MA) studies.

In processes underlying loss of heterozygosity events

Discussing potential molecular pathways or genetic factors that regulate LOH frequency or distribution could deepen the mechanistic understanding of LOH events. For instance, how do chromatin structure, epigenetic modifications, transcriptional activity, or CRISPR-Cas affect LOH rates? Besides HR, NHEJ and BIR.

LOH mediated phenotypic variation.

It briefly mentions LOH in fungal pathogens like *Candida albicans*. A more in-depth discussion of how LOH drives phenotypic variation, particularly in the context of drug resistance and immune evasion, would be worthwhile. For example, the review discusses some studies that have found that drug-resistant strains often exhibit frequent LOH events at specific loci associated with resistance genes. But they don't explain what kind of mutations create homozygous regions, like those that alter drug efflux pumps or reduce the drug's effectiveness, that can become fixed in the population.

The authors briefly discuss LOH in polyploids, but the manuscript would benefit from a section on how LOH operates in polyploid genomes. Some studies have shown differing LOH frequencies and impacts depending on organismal complexity, which could be relevant.

Consider adding to the discussion the latest advances in detecting LOH events, such as new high-throughput sequencing technologies, bioinformatics tools, and their limitations. Given the rapid evolution of genomic technologies, this would provide a practical aspect for readers actively researching LOH in diverse organisms.

In perspectives

The review touches on the role of LOH in cancer but could expand more on how LOH contributes to tumorigenesis. Here, the authors could add the LOH in polyploids.

While the manuscript talks about the significance of LOH, it could end with a more explicit call for future research directions. Highlighting knowledge gaps in understanding LOH, especially about environmental stress and its contributions to species adaptation.

Referee #2:

Page 1: The second paragraph of the introduction seems a little out of place, specifically when it discusses variation in mismatch repair across the yeast genome and cis and trans factors effecting mutation rates. As it is written it is a cursory discussion of a related, but really separate topic. I recommend getting to LOH right away. Otherwise, you will need to expand this section.

Page 5: Burke 2014 and McDonald 2016 are laboratory evolution experiments not mutation accumulation, so are confounded by selection.

Page 6: The statement "most mutation accumulation studies have focused on *S. cerevisiae*" might offend bacterial folks, as a lot of MA studies have been conducted in prokaryotic systems.

Page 6: You mention the *Daphnia pulex* MA experiment (Flynn 2017). I recommend getting rid of this. It doesn't add much and it seems strange to make just this one comparison. Once you start looking beyond *Saccharomyces*, you become obliged to present a more complete comparison, and there have been a lot of MA experiments in metazoans (see PMID: 30476040)

Page 7: Recessive lethal/deleterious mutations are not the only ones that suppress LOH; overdominant mutations do as well (PMIDs: 27194750 & 34363476).

Page 7: In the discussion of Johnson 2021, I don't understand what is meant by "... and homozygous diploids" in the sentence "high-impact mutations in essential genes arose and remained heterozygous in diploids, whereas such mutations were nearly absent in haploids and homozygous diploids."

Page 7: I do not agree with some of discussion regarding hybrids, specifically the comment "New hybrids often experience reduced fitness due to genetic incompatibilities between parental alleles, as several allelic combinations may not have been optimized by natural selection (Kondrashov et al, 2002, Greig et al, 2002; Morales & Dujon, 2012)" The Greig paper examines F1 haploid (and self-mated homozygous diploids). The Morales and Dujon paper examines natural hybrids that have not undergone meiosis and states that "hybrids often exhibit more robust characteristics than the parental strains". Furthermore, I would not characterize the LOH events in the Payen (2016), Lancaster (2019), and Smukowski Heil (2017) papers as "pruning incompatible allelic combinations". Rather these LOH events are better described as revealing recessive or partially dominant beneficial mutations, similar to what was described in the Gerstein (2014) paper mentioned earlier.

Figure 1: Figure 1A doesn't really add anything; I suggest getting rid of it. Adding rate estimates or other quantitative data to Figure 1B would be a useful addition.

Figure 2: Use consistent terminology between Figures 1 and 2. Indicate either on the figure or in the legend that these are all CN-LOH events.

Figure 3: Resolution is low, but this is an easy fix.

Referee #3:

The review is well written and covers most of the LOH mechanisms. If the authors goal is to focus solely on LOH resulting from repair mechanisms then they need to specify this at the beginning. Otherwise, whole chromosome LOH, which arises after a chromosome non-disjunction event, is not covered and should be added for completeness to both, the text and the figures.

Referee#1:

Loss of Heterozygosity events have been a research focus for many decades, and their existence was hypothesized before their observation. Recently, there is increased interest in LOH events, in part because of their importance in experimental evolution studies such as those discussed in the review. The authors nicely connect mechanistic understanding of the LOH events with observations of their occurrence and evolutionary consequences. Other recent reviews have also noted these connections as well (see <https://www.ncbi.nlm.nih.gov/pmc/articles/PMC10276065/>), albeit with a somewhat different focus and some different references. This review could potentially profit from incorporating some references. In particular, additional studies of adaptation following LOH would strengthen the punch line of the paper. I am somewhat concerned about the readability of some of the sections. In the section "Processes underlying loss of heterozygosity events", the second paragraph continues for a long time without breaks. This section needs to be broken up into more digestible chunks (both in paragraph structure and flow).

We appreciate the reviewer's positive comments on our manuscript. As suggested, we have incorporated several relevant references. Additionally, we have expanded the text to include additional examples highlighting the role of loss of heterozygosity in phenotypic variation and adaptation. We also have reorganized the section - "Processes underlying loss of heterozygosity events" - by dividing it into subsections based on the mechanisms and pathways involved, as suggested.

The authors provide a review of LOH mechanisms, including homologous recombination (HR), non-homologous end joining (NHEJ), and break-induced replication (BIR). It also incorporates few experimental evolution studies, natural population genomics surveys, and laboratory-controlled mutation accumulation studies. In processes underlying loss of heterozygosity events Discussing potential molecular pathways or genetic factors that regulate LOH frequency or distribution could deepen the mechanistic understanding of LOH events. For instance, how do chromatin structure, epigenetic modifications, transcriptional activity, or CRISPR-Cas affect LOH rates? Besides HR, NHEJ and BIR.

LOH events occur mechanistically as a result of repair and recombination processes. Accordingly, we have updated the "Mechanisms Driving Loss of Heterozygosity Events" section. Factors such as chromatin structure, epigenetic modifications, and transcriptional activity are not directly involved in these processes. However, we have added text in the manuscript discussing their minor roles in influencing LOH.

LOH mediated phenotypic variation. It briefly mentions LOH in fungal pathogens like *Candida albicans*. A more in-depth discussion of how LOH drives phenotypic variation, particularly in the context of drug resistance and immune evasion, would be worthwhile. For example, the review discusses some studies that have found that drug-resistant strains often exhibit frequent LOH events at specific loci associated with resistance genes. But they don't explain what kind of mutations create homozygous regions, like those that alter drug efflux pumps or reduce the drug's effectiveness, that can become fixed in the population.

We believe that adding specific examples of genes or processes does not significantly enhance the context, as mutations like *MDR1* and *ERG11*, which confer drug resistance in *Candida*, are already extensively documented in the literature. That said, we have included some relevant examples in the text. Additionally, we have incorporated more examples of LOH contributing to pathogenicity in a wider range of organisms. Finally, we have expanded the "LOH mediated phenotypic variation" section to further explain how such mutations arise and become fixed in populations.

The authors briefly discuss LOH in polyploids, but the manuscript would benefit from a section on how LOH operates in polyploid genomes. Some studies have shown differing LOH frequencies and impacts depending on organismal complexity, which could be relevant.

We have added an entire section on LOH in polyploids.

Consider adding to the discussion the latest advances in detecting LOH events, such as new high-throughput sequencing technologies, bioinformatics tools, and their limitations. Given the rapid evolution of genomic technologies, this would provide a practical aspect for readers actively researching LOH in diverse organisms.

We believe that discussing methodologies and tools is beyond the scope of this review, as they have already been comprehensively covered in Heil et al. 2023, as previously mentioned by the reviewer.

In perspectives, The review touches on the role of LOH in cancer but could expand more on how LOH contributes to tumorigenesis. Here, the authors could add the LOH in polyploids. While the manuscript talks about the significance of LOH, it could end with a more explicit call for future research directions.

Highlighting knowledge gaps in understanding LOH, especially about environmental stress and its contributions to species adaptation.

We have added new section in the manuscripts discussing LOH in the polyploid and cancer context. In addition, We do discuss role of LOH in stress and adaptation as well as we have also added future open questions and research.

We have added a new section to the manuscript that discusses LOH in the context of polyploidy and cancer. Additionally, we have addressed the role of LOH in stress response and adaptation. We also included a section highlighting open questions and potential directions for future research.

Referee#2:

Page 1: The second paragraph of the introduction seems a little out of place, specifically when it discusses variation in mismatch repair across the yeast genome and cis and trans factors effecting mutation rates. As it is written it is a cursory discussion of a related, but really separate topic. I recommend getting to LOH right away. Otherwise, you will need to expand this section.

Although we agree that this section of the introduction is not directly related to the main theme of the manuscript, we believe it is crucial in emphasizing that LOH events are one of many types of mutations that significantly influence the evolution and stability of the genome. This paragraph serves to emphasize the importance of LOH relative to other mutations, highlighting that LOH events, due to their significantly higher frequency, deserve greater attention in discussions of genome dynamics.

Page 5: Burke 2014 and McDonald 2016 are laboratory evolution experiments not mutation accumulation, so are confounded by selection.

Thanks for pointing this out. We have corrected this in the text.

Page 6: The statement "most mutation accumulation studies have focused on *S. cerevisiae*" might offend bacterial folks, as a lot of MA studies have been conducted in prokaryotic systems.

We agree and have rephrased the sentence.

Page 6: You mention the *Daphnia pulex* MA experiment (Flynn 2017). I recommend getting rid of this. It doesn't add much, and it seems strange to make just this one comparison. Once you start looking beyond *Saccharomyces*, you become obliged to present a more complete comparison, and there have been a lot of MA experiments in metazoans (see PMID: 30476040).

We have removed the reference to the *Daphnia pulex* study as recommended.

Page 7: Recessive lethal/deleterious mutations are not the only ones that suppress LOH; over dominant mutations do as well (PMIDs: 27194750 & 34363476).

This comment is not clear. However, what we mean by this is “LOH enables organisms to rapidly convert heterozygous regions to homozygosity, accelerating the fixation of advantageous alleles within a population. This process allows beneficial mutations, even those initially recessive, to act as if they were dominant.”

Page 7: In the discussion of Johnson 2021, I don't understand what is meant by "... and homozygous diploids" in the sentence "high-impact mutations in essential genes arose and remained heterozygous in diploids, whereas such mutations were nearly absent in haploids and homozygous diploids."

Thanks for pointing this out. We agree this statement was not accurate and we have rephrased the text as “In diploids, many high-impact mutations in essential genes are fixed as heterozygous, while such mutations are rare in haploids and in homozygous diploids. The accumulation of recessive deleterious mutations in diploids, though expected, poses a challenge in the context of widespread LOH.”

Page 7: I do not agree with some of discussion regarding hybrids, specifically the comment "New hybrids often experience reduced fitness due to genetic incompatibilities between parental alleles, as several allelic combinations may not have been optimized by natural selection (Kondrashov et al, 2002, Greig et al, 2002; Morales & Dujon, 2012)" The Greig paper examines F1 haploid (and self-mated homozygous diploids). The Morales and Dujon paper examines natural hybrids that have not undergone meiosis and states that "hybrids often exhibit more robust characteristics than the parental strains".

We disagree, both the Greig et al. and Morales and Dujon articles comment on the parental incompatibilities in the inter-specific hybrids, which are alleviated by various mechanisms one of which is LOH.

Furthermore, I would not characterize the LOH events in the Payen (2016), Lancaster (2019), and Smukowski Heil (2017) papers as "purging incompatible allelic combinations". Rather these LOH events are better described as revealing recessive or partially dominant beneficial mutations, similar to what was described in the Gerstein (2014) paper mentioned earlier.

We agree and have updated the references to these articles to support the fact stated by Dunham et al. 2002; Dunn et al. 2013; Wolfe 2015.

Referee#3:

The review is well written and covers most of the LOH mechanisms. If the authors goal is to focus solely on LOH resulting from repair mechanisms, then they need to specify this at the beginning. Otherwise, whole chromosome LOH, which arises after a chromosome non-disjunction event, is not covered and should be added for completeness to both, the text and the figures.

We appreciate the reviewer's positive comments on our manuscript. We've improved and added more text to incorporate all possible mechanics, as the reviewer pointed out.

Dr. Joseph Schacherer
Université de Strasbourg
4 rue Konrad Roentgen
Strasbourg 67000
France

Dear Joseph,

I am pleased to inform you that your Review has been accepted for publication in EMBO reports. Your manuscript will be processed for publication by EMBO Press. It will be copy edited and you will receive page proofs prior to publication.

You will soon be contacted by Springer Nature to sign your publishing license. When you login to the customer service website, please use the following token to waive the article publication charges: LTIYMTMZNTUXOA

Should you experience any difficulty, please email publishing@embo.org.

If you have any questions, please do not hesitate to contact the Editorial Office. Thank you very much for your contribution to EMBO Reports.
